# MAPO: MIXED ADVANTAGE POLICY OPTIMIZATION

## ABSTRACT

Recent advances in reinforcement learning for foundation models, such as Group Relative Policy Optimization (GRPO), have significantly improved the performance of foundation models on reasoning tasks. Notably, the advantage function serves as a central mechanism in GRPO for ranking the trajectory importance. However, existing explorations encounter both advantage reversion and advantage mirror problems, which hinder the reasonable advantage allocation across different query samples. In this work, we propose an easy but effective GRPO strategy, **M**ixed **A**dvantage **P**olicy **O**ptimization (**MAPO**). We reveal that the trajectory appears with different certainty and propose the advantage percent deviation for samples with high-certainty trajectories. Furthermore, we dynamically reweight the advantage function for samples with varying trajectory certainty, thereby adaptively configuring the advantage function to account for sample-specific characteristics. Comparison with related state-of-the-art methods, along with ablation studies on different advantage variants, validates the effectiveness of our approach.

## 1 INTRODUCTION

Recent advances in the reasoning capabilities of Foundation Model (FM) Jaech et al. (2024); Team et al. (2025); Guo et al. (2025); Wen et al. (2025); Guo et al. (2025); Wang et al. (2025b) have been largely driven by improvements in long Chain of Thought (CoT) generation. Among various enhancement strategies, Reinforcement Learning (RL) Ouyang et al. (2022); OpenAI (2023); Shao et al. (2024); Hu et al. (2025); Xiong et al. (2025) has emerged as a powerful post-training technique, enabling FM to refine their CoT reasoning through self-improvement. Therefore, RL serves as the key mechanism for unlocking the reasoning ability in various domains.

Notably, Group Relative Policy Optimization (GRPO) Shao et al. (2024) is introduced as a popular reinforcement strategy. GRPO generates and refines a group of reasoning paths through the group-relative advantage estimation based on rule-based reward functions. Thus, a key difference with traditional reinforcement methods, such as proximal policy optimization Schulman et al. (2017) and direct preference optimization Rafailov et al. (2023); Chen et al. (2024); Liu et al. (2025a), is that GRPO eliminates the need for an additional learned reward critic model, instead leveraging efficient sampling from the Foundation Model policy. Witnessing the success of Group Relative Policy Optimization, its advantage function plays a key role in promoting trajectories with relatively higher advantages, thereby guiding the policy model to update towards more reliable directions. Despite recent advancements, GRPO and its variants generally maintain a fixed advantage formulation throughout the entire training cycle Guo et al. (2025); Yao et al. (2025); Guo et al. (2025). However, this approach overlooks a significant challenge: *the fixed advantage fails to provide meaningful signals for samples with varying trajectory certainty degrees*.

To analyze the drawbacks of existing advantage formulations, we first define ***trajectory certainty*** within the sampling group. The advantage is computed from verifiable rewards, typically format and accuracy metrics, to jointly measure the trajectory score Guo et al. (2025); Shao et al. (2024); Yao et al. (2025); Zhang et al. (2025); Liu et al. (2025b); Xu & Ding (2025). For a sampled trajectory, we declare the **success** only if it achieves the correct answer on all reward metrics. Consequently, each trajectory can be viewed as a Bernoulli trial with outcome: failure or success. Then, in the group sampling, the number of successes over repeated draws follows a binomial distribution, and *high-certainty samples tend to yield nearly identical outcomes across draws*, *i.e.*, samples that are too hard or too easy. We then formally derive the definition of trajectory certainty as follows:

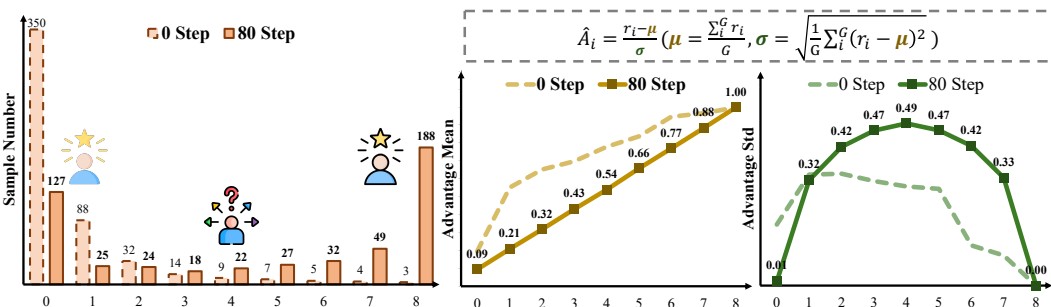

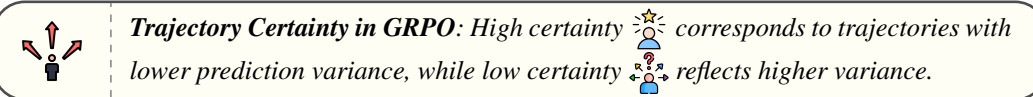

Figure 1: **Observation**. During the reinforcement, different samples appear with diverse successful trajectory numbers $N = \sum_{i=1}^{G} \mathbf{1}_{\{r_i=1\}}$ (**X-axis**). Samples with lowest trajectory certainty tends to achieve most diverse prediction pattern, *i.e.*, $N = 4$. Experiments are conducted on the Geo3K with rollout number $G = 8$.

> ***Trajectory Certainty in GRPO***: *High certainty corresponds to trajectories with lower prediction variance, while low certainty reflects higher variance.*

We analyze the sample behavior in fig. 1 and reveal two underlying limitations of the existing advantage paradigm. *First*, **Advantage Reversion**: high-certainty samples may receive more differentiated advantage allocations than low-certainty ones. Specifically, a high-certainty sample with rewards $r_{\text{High}} = \{0.9, 1.0, 1.0, 1.0\}$ receives a more discriminative advantage allocation than a low-certainty sample with $r_{\text{Low}} = \{0.1, 0.9, 1.0, 1.0\}$ due to **a small advantage standard deviation** $\sigma$. However, high-certainty samples do not require strong penalization, whereas low-certainty trajectories benefit from stronger correction. *Second*, **Advantage Mirror**: high-certainty samples ( & ) also require distinct advantage allocation for extreme cases. In particular, the existing advantage formulation does not take into account the **monotonic advantage scores** $\mu$ and therefore treats easy and hard samples indistinguishably. The core issue is that the same advantage formulation cannot be applied uniformly across samples with different trajectory certainty. In summary, this motivates us to rethink the advantage design and decomposes it into two sub-questions: **i)** *how to design the advantage function for high-certainty samples*? and **ii)** *how to adaptively combine advantage functions for samples with varying trajectory certainty*?

To address the question **i)**, we introduce the Advantage Percent Deviation (APD), which replaces the advantage from standard z-score normalization to relative normalization. Specifically, the original advantage formulation is expressed as $\hat{A}_i = \frac{r_i - \mu}{\sigma}$. For high-certainty sample trajectories, this formulation fails to capture the overall level of reward scores. Besides, variance in the rollout trajectory can yield a small $\sigma = \text{std}(\boldsymbol{r})$, which in turn leads to numerical instability and uncontrollable boundary on advantage allocation. To deal with this drawback, we introduce a novel advantage function for high-certainty samples as $\hat{A}_i^{APD} = \frac{r_i - \mu}{\mu}$. Regarding question **ii)**, we propose the Trajectory Certainty Reweight (TCR) to determine the sample advantage function based on trajectory certainty. Inspired by Bernoulli sampling, each trajectory is treated as either a success or a failure. A trajectory group exhibits the highest uncertainty when the success-to-failure ratio is fifty percent. Therefore, we use trajectory certainty to dynamically reweight the advantage function from $\hat{A}_i$ to $\hat{A}_i^{APD}$. In this work, we argue that the existing advantage formulation is not consistently appropriate for samples with varying levels of trajectory certainty. To address this issue, we propose a simple yet effective method, **M**ixed **A**dvantage **P**olicy **O**ptimization (**MAPO**), which rethinks the advantage formulation in GRPO. To validate our approach, we conduct extensive experiments across multiple datasets using the Qwen2.5-VL-7B architecture, demonstrating the superior performance in both In-Domain and Out-of-Domain aspects. Our contributions are summarized as follows:

- We focus on the Group Relative Policy Optimization paradigm and reveal that existing advantage formulation faces two unavoidable challenges: advantage reversion and advantage mirror.
- We propose Mixed Advantage Policy Optimization (MAPO), a simple yet effective method to overcome existing advantage limitations. Preliminary, we introduce trajectory certainty to evaluate sample behavior. We propose Advantage Percent Deviation for high-certainty advantage estimation and utilize Trajectory Certainty Reweight to dynamically construct the advantage function.

- We perform a comprehensive analysis on reasoning scenarios, including mathematics and emotion fields. Through a series of ablation studies, the promising results empirically validate the effectiveness of the proposed mixture advantage strategies in enhancing GRPO overall performance.

## 2 RELATED WORKS

### 2.1 FOUNDATION MODEL

The development of Large Language Model (LLM) has revolutionized artificial intelligence, significantly transforming the way machines understand and generate human language. Notable examples of LLM include the GPT series Radford et al. (2019); Brown et al. (2020); OpenAI (2023), Meta LLaMA Touvron et al. (2023), and Google PaLM Chowdhery et al. (2022); Anil et al. (2023), all of which have demonstrated impressive capabilities in natural language understanding and generation. These advancements have sparked considerable interest in extending LLM to handle multi-modal inputs, particularly by incorporating vision components, which has led to the development of Multimodal Large Language Model (MLLM). Building on the success of LLM, growing interest has emerged in constructing end-to-end Multimodal Large Language Model (MLLM) systems, such as Flamingo Alayrac et al. (2022), BLIP-2 Li et al. (2022; 2023), InstructBLIP Dai et al. (2023), QWen-VL Bai et al. (2023b), LLaVA Liu et al. (2023b;a); Zhu et al. (2024); Li et al. (2024), and VILA Lin et al. (2023); Fang et al. (2024); Liu et al. (2025d). Existing MLLM solutions typically rely on visual extractors Radford et al. (2021); Dosovitskiy et al. (2021); Caron et al. (2021) to encode visual features, using a connector module to project visual tokens into the word embedding space of the LLM, *i.e.*, treating visual input as a foreign language Wang et al. (2023). Subsequently, the visual and textual tokens are concatenated and fed into the LLM. The LLM is then used to perform various vision-language tasks in an auto-regressive manner. As a result, foundation models are gradually evolving from a single-textual modality to multimodal capabilities. However, existing works predominantly focus on supervised fine-tuning (SFT) on large-scale pre-training datasets. The success of OpenAI o1 Radford et al. (2019); Jaech et al. (2024); Ouyang et al. (2022) highlights the powerful potential of reinforcement learning in post-training to enhance model reasoning capabilities. With the open-sourcing of Deepseek-R1 Guo et al. (2025) and Qwen Bai et al. (2023a;b); Yang et al. (2025b;a), reasoning models are now widely deployed locally, drawing attention from the research community to the efficiency of long chain-of-thought generation for foundation models. And utilizing the reinforcement technique to empower foundation models with reasoning capabilities has emerged as a pivotal methodology beyond the limitations of SFT.

### 2.2 GROUP RELATIVE POLICY OPTIMIZATION

Several methods have been proposed to elicit reasoning abilities on mathematical and scientific problems, enabling foundation models to better handle inference and analysis. Especially, Group Relative Policy Optimization (GRPO) has recently garnered significant attention in the research field, as its rule-based reward function effectively enhances the reasoning capabilities of large models. Existing exploration or variants of GRPO could normally be divided into the following streams. ❶ *Think Trajectory Diversity*. This paradigm focuses on diversifying the thinking process to facilitate a more meaningful candidate rollout. Specifically, it boosts the trajectory from two angles: input perturbation and process polish. First, constructing the data augmentation technique for Multimodal Large Language Model to enhance both the quantity and quality of training data. NoisyRollout Liu et al. (2025b) leverages the noise annealing schedule to construct the noisy image text pairs. VP Li et al. (2025) introduces three targeted perturbations: distractor concatenation, dominance-preserving mixup, and random rotation. Share-GRPO Yao et al. (2025) turns to expand the question space for a given question via data transformation. Second, polishing the thinking process acts as a reliable direction to monitor the thinking behavior. StepGRPO Zhang et al. (2025) requires the think process to explicitly reveal key intermediate steps. Both SophiaVL-R1 Fan et al. (2025) and GRPO-CARE Chen et al. (2025b) utilize an external thinking reward model that evaluates the quality of the entire thinking process. MGRPO Ding et al. (2025) recycles previous think messages for self-correction learning. Hint-GRPO Huang et al. (2025) adaptively provides hints to the samples. However, the aforementioned solutions require constructing dedicated data augmentation strategies or modifying the thinking process, which introduces additional computational costs or an external thinking reward evaluation model. ❷ *Reward Formulation Refinement*. With respect to verifiable reward

Table 1: **Weakness** for different GRPO variants. Refer to Sec. 2.2 for details.

| Methods | Input Space Augmentation | Think Cost Increase | Specific Task Adaption | Additional Hyper-Parameter |
|---|---|---|---|---|
| *Think Trajectory Diversity* 🏆 | | | | |
| NoisyRollout | ✓ (Noisy Distortion) | | | ✓ (Initial Noise Strength) |
| VP | ✓ (Visual Augmentation) | | | ✓ (Perturbation Types) |
| Share-GRPO | ✓ (Textual Enrichment) | | | ✓ (Question Variants Number) |
| StepGRPO | | ✓ (Step Think) | | ✓ (Key Steps Number) |
| GRPO-CARE | | ✓ (Reference Model) | | ✓ (Consistency Coefficient) |
| *Reward Formulation Refinement* 🎯 | | | | |
| Visual-RFT | | | ✓ (Visual IoU Reward) | |
| GRPO-$\lambda$ | | | ✓ (Length Penalty) | ✓ (Top-$\lambda$ Fraction) |
| GRPO-LEAD | | | ✓ (Length Reward) | ✓ (Advantage Rescale Factor) |
| *Advantage Estimation Redesign* 📊 | | | | |
| SEED-GRPO | | | | ✓ (Advantage Rescale Factor) |
| GPG | | | | ✓ (Valid Sample Threshold) |

construction Bi et al. (2024); Team et al. (2025), it is the predefined rules and normally incorporates the Format Reward and Accuracy Reward. The former requires the model output should meet the required HTML tag format of <think> and <answer>. The latter is determined by comparing the model output class with the ground truth class, yielding a value of 1 for correct classification and 0 for incorrect classification. Thus, recent works design different verifiable reward functions for different specific tasks. For instance, Visual-RFT Liu et al. (2025f) proposes the intersection over union reward for object detection. VisionReasoner Liu et al. (2025c) introduces diverse perception rewards in a unified framework. Both GRPO-$\lambda$ Dai et al. (2025a) and GRPO-LEAD Zhang & Zuo (2025) consider the over-length penalty reward. However, this pattern typically focuses on adapting to specific tasks, which limits cross-task generalization. Additionally, it requires careful tuning of hyperparameter reward weights for different reward metrics. Therefore, this pattern fails to achieve robust performance across diverse real-world application settings. ❸ *Advantage Estimation Redesign*. Towards advantage estimation, recent researches investigate better trajectory importance measurement via reformulation or rescaling operation. Dr. GRPO Liu et al. (2025e) and GPG Chu et al. (2025b) consider removing the standard deviation to alleviate the reward bias. SEED-GRPO Chen et al. (2025a) reweights the advantages based on the semantic entropy to measure the output uncertainty. KRPO Wang et al. (2025a) introduces a lightweight Kalman filter approach for accurate advantage estimation. But this paradigm faces the hyperparameter selection dilemma, or consistent advantages for different samples. We conclude the weakness of existing GRPO variants in Tab. 1. In our work, we reveal that *samples appear distinct trajectory certainty behavior and utilizing uniform advantage strategy unavoidably degrades partial samples optimization*. Therefore, we dynamically set the advantage function based on the trajectory certainty to boost the overall reinforcement effect.

## 3 METHODOLOGY

### 3.1 PRELIMINARY

Group Relative Policy Optimization (GRPO) Shao et al. (2024) is a variant of Proximal Policy Optimization (PPO) Schulman et al. (2017) originally developed to enhance mathematical reasoning in LLM. However, it can also be effectively adapted to improve visual reasoning in Multimodal Large Language Model. GRPO begins by constructing the current policy model $\pi_\theta$ and a reference model $\pi_{old}$, where the latter represents the "old" policy or the policy from a previous iteration. Let $\rho_Q$ denote the distribution of prompts or questions. Given a prompt $q \sim \rho_Q$, GRPO samples a group of outputs $o_1, o_2, \ldots, o_G$ from the old model $\pi_{old}$. It then optimizes the policy model $\pi_\theta$ by maximizing the following objective function:

$$\mathcal{J}_{\text{GRPO}}(\theta) = \mathbb{E}_{q \sim \rho_Q} \mathbb{E}_{o \sim \pi_{old}(\cdot|q)} \left[ \frac{1}{G} \sum_i^G f_\epsilon \left( \frac{\pi_\theta(o_i|q)}{\pi_{old}(o_i|q)}, \hat{A}_i \right) \right] - \beta \mathbb{D}_{KL}[\pi_\theta || \pi_{ref}], \quad (1)$$

where $\beta$ is the hyper-parameter. $f_\epsilon(x, y) = \min(xy, \text{clip}(x, 1 - \epsilon, 1 + \epsilon)y)$. $\hat{A}_i$ is the advantage calculated based on the relative rewards of the outputs inside each group. To be precise, for each question $q$, a group of outputs $\{o_1, o_2, \ldots, o_G\}$ are sampled from the old policy model $\pi_{old}$. A reward function ($R$) is then used to score the outputs, yielding $G$ rewards $\boldsymbol{r} = \{r_1, r_2, \ldots, r_G\}$

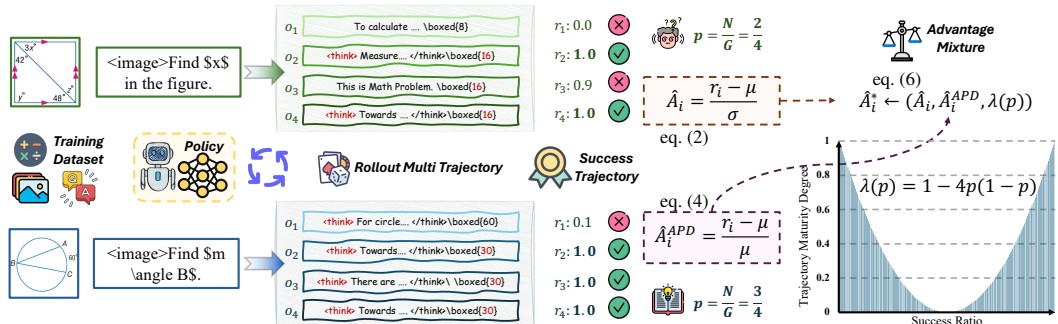

Figure 2: **Architecture illustration of MAPO**. We reveal that the trajectory certainty varies across samples. In general, we introduce the Advantage Percent Deviation to replace the advantage function for high-certainty elements. We utilize the Trajectory Certainty Reweight to dynamically reweight the advantage function via trajectory certainty. Assume rollout number $G=4$. Best viewed in color. Zoom in for details. See Sec. 3.2.

correspondingly, where $r_i = R(q, o_i)$. The mean reward is then calculated as $\mu = \frac{1}{G} \sum_{i=1}^{G} r_i$ and the standard deviation is defined as $\sigma = \sqrt{\frac{1}{G} \sum_{i=1}^{G} (r_i - \mu)^2}$. The default normalized advantage for the $i^{th}$ rollout is defined as the following formulation:

$$\hat{A}_i = \frac{r_i - \mu}{\sigma}. \tag{2}$$

## 3.2 PROPOSED METHOD

**Observation**. We model the trajectory outcome as a Bernoulli random variable, $X \sim$ Bernoulli$(p)$, $X \in \{0, 1\}$, where $X = 1$ denotes a successful trajectory and $X = 0$ denotes a failure. The success probability $p$ is defined by the expectation of $X$, $\mathbb{E}[X] = p$, and the variance of this distribution is $\text{Var}(X) = p(1 - p)$, which quantifies the certainty of the trajectory outcome, shown in fig. 1. However, directly measuring $p$ is challenging, so we estimate it empirically using the ratio $p \approx \frac{N}{G}$, where $G$ is the total number of sampled trajectories. $N$ is the number of successful trajectories and is defined as the following formulation:

$$N = \sum_{i=1}^{G} \mathbf{1}_{\{r_i=1\}}. \tag{3}$$

This empirical estimation approximates the true probability $p$ via observed trajectories. Thus, we reveal that *samples exhibit varying certainty level within the GRPO sampling process*.

**Advantage Percent Deviation**. Sample would appear high certainty, when the prediction variance is close to zero ($\text{Var}(X) \to 0$), *i.e.*, $p \to 0$ or $p \to 1$, which typically corresponds to overly easy or difficult instances. In such cases, existing advantage faces two key challenges: *Advantage Reversion* and *Advantage Mirror*. Specifically, the advantage formulation, $\hat{A}_i = \frac{r_i - \mu}{\sigma}$, can produce misleading behaviors between trajectories with high and low certainty. For instance, in the case of *Advantage Reversion*, the high-certain trajectory with a relatively high reward of 0.9 in a batch of $\boldsymbol{r}_{\text{High}} = [0.9, 1, 1, 1]$ is assigned a large negative advantage ($\min \hat{A}_i = -1.73$), which is more extreme than the low-certain ones like $\boldsymbol{r}_{\text{Low}} = [0.1, 0.1, 1, 1]$, due to the small standard deviation exaggerating deviations from the mean. Similarly, as for *Advantage Mirror*, two reward batches that are symmetric around the center, such as $[0, 0.1, 0.1, 0.1]$ and $[0.9, 1, 1, 1]$, yield mirrored normalized advantage scores $[-1.73, 0.57, 0.57, 0.57]$, which makes semantically distinct cases to appear structurally equivalent normalization. These examples show how reliance on $\mu$ and $\sigma$ alone can distort the relative evaluation of trajectories, especially when the variance is abnormally small or the rewards are symmetrically distributed, thus echoing a more robust advantage function.

Therefore, in our work, to address the question **i)**: *high-certainty samples advantage reconstruction*, we introduce the Advantage Percent Deviation (APD), to effectively address the issues of *Advantage Reversion* and *Advantage Mirror*. Instead of relying on z-score normalization, APD measures the relative deviation of each trajectory reward from the batch mean reward, formulated as follows:

$$\hat{A}_i^{APD} = \frac{r_i - \mu}{\mu}. \tag{4}$$

| Methods | Advantage Formulation | Advantage Problem | |
|---|---|---|---|
| | | Adv. Reversion | Adv. Mirror |
| GRPO ♣ | $\hat{A}_i = \frac{r_i - \mu}{\sigma}$ | ✓ | ✓ |
| Dr. GRPO ♦ | $\hat{A}_i = r_i - \mu$ | | ✓ |
| GPG ♥ | $\hat{A}_i = \alpha * (r_i - \mu)$ | | ✓ |
| TreeRPO ♠ | $\hat{A}_i = \frac{r_i - \mu}{\mu(1-\mu)}$ | | ✓ |
| MAPO ★ | $\hat{A}_i = (1 - \lambda(p))\frac{r_i - \mu}{\sigma} + \lambda(p)\frac{r_i - \mu}{\mu}$ | - | - |

Figure 3: **Discussion on existed advantage functions**. $r$ means the group reward. We denote $\mu = mean(r)$ and $\sigma = std(r)$. The reward is defined as a combination of the format reward ($r_{Format}$) and the accuracy reward ($r_{Accuracy}$), with a weighting factor of $\beta = 0.9$, *i.e.*, $r = (1 - \beta)r_{Format} + \beta r_{Accuracy}$. $\alpha = 0.6$ is the valid sample rescale parameter for GPG Chu et al. (2025b). Refer to Sec. 3.3 for details.

This design emphasizes the proportional difference between individual rewards and the central tendency, ensuring that the advantage reflects not only the relative ordering but also the magnitude of deviation in percentage terms. By doing so, APD mitigates the instability caused by abnormally small standard deviations and prevents mirrored advantage allocation from being treated as equivalent, thereby providing a more stable and reasonable trajectory quality evaluation.

**Trajectory Certainty Reweight**. Considering that samples are in various trajectory certainty conditions, it is essential to dynamically adjust the advantage formulation across different samples. We propose the Trajectory Certainty Reweight (TCR), which adaptively reconstructs the advantage function based on trajectory certainty to address **ii)** *various-certainty sample advantage reweighting*. This design ensures that sample-specific characteristics are preserved, leading to a more faithful and stable evaluation of trajectory quality.

To be precise, we formalize TCR by introducing a certainty-aware weighting scheme. The key intuition is that when a trajectory exhibits high uncertainty (immature stage), the advantage should rely more on a variance-sensitive formulation (eq. (2)), while in highly certain (mature) stages it should instead emphasize a mean-relative formulation (eq. (4)) that remains stable even when variance collapses. To operationalize this idea, we use the estimated trajectory certainty $p$ to interpolate these two advantages for different samples. We denote the trajectory certainty degree as follows:

$$\lambda(p) = 1 - 4p(1 - p) \in [0, 1] \quad (p \in [0, 1]). \tag{5}$$

And, then we further construct the sample-wise advantage construction as follows:

$$\hat{A}_i^* = (1 - \lambda(p)) * \underbrace{\frac{r_i - \mu}{\sigma}}_{\textbf{Deviation-based}} + \lambda(p) * \underbrace{\frac{r_i - \mu}{\mu}}_{\textbf{Mean-based}}. \tag{6}$$

The standard deviation–based advantage is weighted by $1 - \lambda(p)$, while the complementary factor $\lambda(p)$ is assigned to the mean-based advantage. In this way, the contribution shifts smoothly from deviation-based signals under uncertainty to mean-based signals under certainty, ensuring a balanced and robust construction of the advantage function across different trajectory certainty levels. Thus, we replace the original advantage $\hat{A}_i$ to proposed $\hat{A}_i^*$ in eq. (6) in eq. (1) for optimization. As a result, our method reveals the trajectory certainty phenomenon and effectively mitigates existing advantage limitations via dynamical advantage reweight operation. We provide the methodological framework in fig. 2 and the algorithm description in algorithm 1.

## 3.3 DISCUSSION AND LIMITATION

**Advantage Exploration**. The advantage function typically relies on group-based estimation from trajectory rewards $r_i \in r$. Existing explorations Liu et al. (2025e); Chu et al. (2025b); Yang et al. (2025c) can be broadly classified as the following directions. **First**, methods such as Dr. GRPO Liu et al. (2025e), GPG Chu et al. (2025b), S-GRPO Dai et al. (2025b), and wd1 Tang et al. (2025) remove the standard variance normalization term to alleviate reward bias. More recently, Yang et al. (2025c) identifies that conventional normalization fails to scale advantages properly under continuous rewards, and proposes to rewrite the variance as $\sigma = \mu(1 - \mu)$ to address this issue. Meanwhile, KRPO Wang et al. (2025a) introduces a lightweight Kalman filter to dynamically estimate latent reward mean and variance, enabling more adaptive advantage normalization. However, these approaches fail to resolve both *advantage reversion* and *advantage mirror* problems in fig. 3.

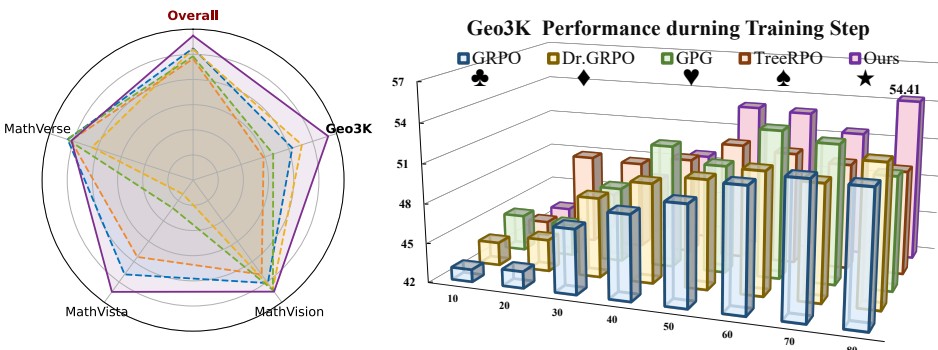

Figure 4: **Performance comparison with different advantage formulations** on the geometry task based on Qwen2.5-VL-7B-Instruct model with rollout number $G=12$. Please refer to Sec. 3.3 for details.

Moreover, existing methods typically employ a uniform advantage formulation across all samples, overlooking the uniqueness of individual sample conditions. In contrast, we observe that samples exhibit varying degrees of certainty during optimization. Motivated by this insight, we propose a novel advantage function that leverages relative deviation for high-certainty samples, and further introduce a certainty-aware reweighting scheme that dynamically adjusts the advantage construction based on trajectory certainty. This design ensures a more faithful and stable evaluation of the sample situation across diverse training conditions. We further conduct the empirical experiments to validate the proposed mixture advantage solutions in fig. 4 and MAPO achieves a satisfying performance.

**Conceptual Difference**. Utilizing GRPO to enhance foundation models with reasoning capabilities has gained significant attention Liu et al. (2025e); Chen et al. (2025b); Chu et al. (2025a); Ma et al. (2025). Recent studies have explored the contribution of sample selection to the effectiveness of GRPO across three main streams. First, one line focuses on highlighting relatively simple samples to achieve a stable optimization. For instance, SEED-GRPO Chen et al. (2025a) utilizes semantic entropy to measure answer diversity and applies more conservative updates to hard questions. However, blindly emphasizing easy samples can restrict the model exploration ability Yue et al. (2025) and face model entropy collapse Zhang et al. (2024a). Second, another paradigm seeks to highlight hard samples. For example, GRPO-LEAD amplifies learning signals for challenging problems using a difficulty-aware advantage reweight. Recently, Pikus et al. (2025) has pointed out that the hardest examples consistently yield superior performance on reasoning benchmarks. However, this approach leads to longer convergence times. The third group focuses on eliminating meaningless samples. DAPO Yu et al. (2025) and GPG Chu et al. (2025b) aim to discard samples with vanishing advantages, *i.e.*, $\sigma = 0$. DAPO considers over-sampling and filtering out prompts with mean accuracy $\mu \in \{0, 1\}$, but this operation is not efficient in terms of training time. This inefficiency arises because the time required to collect a batch of desired examples is uncontrollable and depends on the task difficulty. In contrast, GPG seeks more accurate gradient estimation by rescaling the gradient based on the valid samples ratio, with a validity threshold of $0.6$. In summary, existing work conducts a **monotonic emphasis** based on sample difficulty, which inevitably faces the prisoner dilemma of sample difficulty. In contrast, our work considers trajectory certainty and allocates different mixture ratios for high- and low-certainty samples, thereby introducing a **discriminative emphasis**. We reveal the gradient of MAPO compared with GRPO. *Without loss of generality*, we simplify the gradient analysis by ignoring clipping and KL regularization and considering the reward as a Bernoulli variable. We define the ratio between the gradients of MAPO and GRPO as:

$$\varrho(p) \triangleq \frac{\nabla_\theta \mathcal{J}_{\mathrm{MAPO}}}{\nabla_\theta \mathcal{J}_{\mathrm{GRPO}}} = (1 - \lambda(p)) + \lambda(p)\sqrt{\frac{1-p}{p}}, \quad \lambda(p) = 1 - 4p(1-p). \quad (7)$$

By further analysis (see details in appendix C), we obtain the following formulation:

$$\begin{cases} \varrho(p) > 1, & p \in (0, \frac{1}{2}), \\ \varrho(p) = 1, & p = \frac{1}{2}, \\ 0 < \varrho(p) < 1, & p \in (\frac{1}{2}, 1). \end{cases} \quad (8)$$

This shows that the mixed reward, MAPO *implicitly assigns larger gradients to harder samples (with $p < \frac{1}{2}$) and smaller gradients to easier ones (with $p > \frac{1}{2}$)*, which aligns with prior insights that appropriately emphasizing difficult samples enhances the performance GRPO Pikus et al. (2025).

Table 2: **Ablative study of key modules** for MAPO viaQwen2.5-VL-7B-Instruct with rollout number $G = 12$. Incorporate **sole** Advantage Percent Deviation (APD) can be regarded as the advantage replacement. Involving **both** APD and TCR achieves a satisfying performance. Please refer to Sec. 4.2. for details.

| APD | TCR | Geo3K | MathVision | MathVista | MathVerse | $\mathcal{A}^{\mathcal{T}}$ | $\bar{\mathcal{A}}$ | EmoSet | WEBEmo | Emotion6 | $\mathcal{A}^{\mathcal{T}}$ | $\bar{\mathcal{A}}$ |
|---|---|---|---|---|---|---|---|---|---|---|---|---|
| Vanilla | | - | - | - | - | - | - | - | - | - | - | - |
| | | 51.91 | 26.74 | 72.70 | 43.93 | 47.79 | 49.85 | 75.50 | 49.90 | 60.44 | 55.17 | 65.33 |
| ✓ | | 50.92 | 26.61 | 73.00 | 44.59 | 48.07 | 49.49 | 77.20 | 50.55 | 61.95 | 56.25 | 66.72 |
| ✓ | Rand | 53.41 | 24.90 | 71.20 | 43.38 | 46.49 | 49.95 | 76.90 | 50.20 | 62.12 | 56.16 | 66.53 |
| ✓ | ✓ | **54.41** | 27.30 | 73.20 | 43.81 | **48.10** | **51.26** | **77.86** | 50.75 | 60.61 | **55.68** | **66.77** |

**Limitation**. Despite achieving satisfactory performance with free hyperparameters, our research has several limitations. First, our approach uses trajectory certainty to treat different samples selectively. In extreme reinforcement scenarios or when foundational model capabilities are limited, it becomes difficult to generate a diverse set of successful trajectories, as rollout may consistently fail. In such cases, our method could reduce to a single function strategy. Second, although our method assigns different reward mechanisms for samples with different trajectory maturity levels, a more refined reward allocation method is still worth exploring. Third, due to computational constraints, our experiments are limited to models with up to 7B parameters and datasets with a few thousand samples. Future work would aim to extend these findings to larger-scale scenarios.

## 4 EXPERIMENTS

### 4.1 EXPERIMENTAL SETUP

**Environment and Datasets**. We conduct experiments on two reasoning scenarios: mathematics and emotion. For training, we utilize Geo3K [arXiv'21] Lu et al. (2021) and EmoSet [ICCV'23] Yang et al. (2023). These two datasets are respectively comprised of $2.1K$ training samples. Furthermore, for the out-of-domain validation, we respectively adopt out-of-domain datasets in the math and emotion fields: MathVista [arXiv'23] Lu et al. (2023), MathVision [NeurIPS'24] Wang et al. (2024), MathVerse [ECCV'24] Zhang et al. (2024b), WEBEmo [ECCV'18] Panda et al. (2018), and Emotion6 [CVPR'15] Peng et al. (2015). We provide a detailed dataset illustration in appendix B.1.

**Architecture and Counterparts**. We utilize the popular open-source Qwen2.5-VL-7B-Instruct as the base (Vanilla) model, which exhibits strong foundational capabilities well-suited for subsequent RL training Yang et al. (2025b); Bai et al. (2025). We further conduct the comparison with the GRPO Shao et al. (2024) and DAPO Yu et al. (2025) to validate the effectiveness of our method.

**Implementation Details**. Experiments are conducted on 8 A100 GPUs. Detail is in appendix B.2.

**Evaluation Metrics**. We evaluate both in-domain ($\mathcal{A}^{\mathcal{S}}$) and out-of-domain ($\mathcal{A}^{\mathcal{T}}$). Let $\mathcal{T} = \{\mathcal{T}_t\}_{t=1}^{|\mathcal{T}|}$ represent the unseen dataset set and $\mathcal{S}$ denote the training distribution. Thus, we derive the following evaluation metrics forms $\mathcal{A}^{\mathcal{S}} = \text{Acc.}(\mathcal{S})$ and $\mathcal{A}^{\mathcal{T}} = \frac{1}{|\mathcal{T}|}\sum_t^{|\mathcal{T}|}\text{Acc.}(\mathcal{T}_i)$. Acc. denotes the accuracy metric. Furthermore, we use the Average metric to evaluate overall performance as $\bar{\mathcal{A}} = \frac{\mathcal{A}^{\mathcal{S}} + \mathcal{A}^{\mathcal{T}}}{2}$.

### 4.2 DIAGNOSTIC ANALYSIS

We perform ablation studies on the Geo3K and EmoSet datasets, utilizing the Qwen2.5-VL-7B-Instruct model to facilitate an in-depth analysis. We quantitatively analyze the proposed Mixed Advantage Policy Optimization (MAPO) in Tab. 2. The ablation demonstrates that solely replacing the advantage function from the original $\hat{A}_i = \frac{r_i - \mu}{\sigma}$ to $\hat{A}_i^{APD} = \frac{r_i - \mu}{\mu}$ leads to limited performance improvement or even degradation, which underscores the necessity for a dynamic advantage function. Furthermore, utilizing random weight allocation ($\lambda(p) \sim \mathcal{U}(0,1)$ in eq. (6)) fails to achieve stable performance improvements. Thus, incorporating the Trajectory Certainty Reweight, which accounts for trajectory certainty, further enhances overall performance.

### 4.3 COMPARISON TO STATE-OF-THE-ARTS

We benchmark MAPO against state-of-the-art reinforcement learning frameworks in reasoning tasks. As illustrated in Tab. 3, MAPO consistently outperforms Vanilla, GRPO, and DAPO across

Table 3: **Performance comparison with GRPO variants** on the geometry and emotional reasoning tasks. We mark the **Best** in bold an Second in underline across different methods. Refer to Sec. 4.3.

| Methods | Geo3K | MathVision | MathVista | MathVerse | $\mathcal{A}^{\mathcal{T}}$ | $\bar{\mathcal{A}}$ | EmoSet | WEBEmo | Emotion6 | $\mathcal{A}^{\mathcal{T}}$ | $\bar{\mathcal{A}}$ |
|---|---|---|---|---|---|---|---|---|---|---|---|
| Vanilla | 37.43 | 24.51 | 67.20 | 40.02 | 43.91 | 40.67 | 53.65 | 46.85 | 52.19 | 49.52 | 51.58 |
| *Qwen2.5-VL-7B-Instruct with Rollout Number $G = 12$* | | | | | | | | | | | |
| GRPO | 51.91 | 26.74 | 72.70 | 43.93 | 47.79 | 49.85 | 77.20 | 49.90 | 60.44 | 55.17 | 66.18 |
| DAPO | 52.91 | 26.51 | 73.50 | 44.59 | **48.20** | 50.56 | 76.05 | 50.60 | 60.61 | 55.60 | 65.82 |
| MAPO | **54.41** | 27.30 | 73.20 | 43.81 | 48.10 | **51.26** | **77.86** | 50.75 | 60.61 | **55.68** | **66.77** |
| *Qwen2.5-VL-7B-Instruct with Rollout Number $G = 8$* | | | | | | | | | | | |
| GRPO | 50.92 | 26.38 | 72.60 | 43.45 | **47.48** | 49.20 | 76.40 | 49.90 | 60.27 | 55.08 | 65.74 |
| DAPO | 50.42 | 26.41 | 72.40 | 43.15 | 47.32 | 48.87 | 68.44 | 47.80 | 58.08 | 52.94 | 60.69 |
| MAPO | **54.24** | 27.37 | 71.30 | 43.40 | 47.36 | **50.80** | **77.46** | 50.05 | 61.28 | **55.66** | **66.56** |

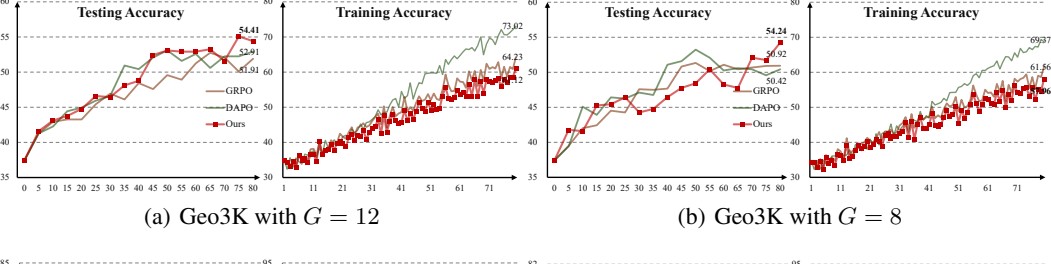

(a) Geo3K with $G = 12$      (b) Geo3K with $G = 8$

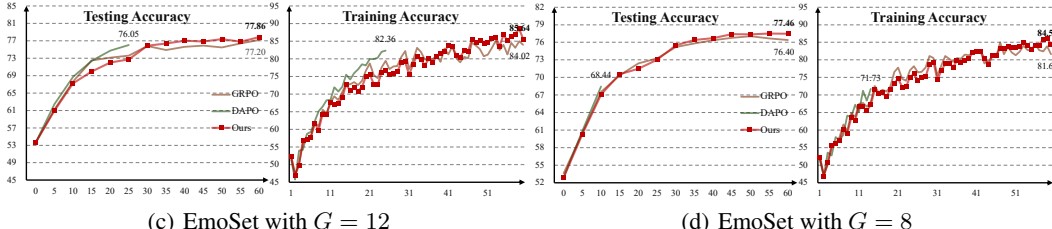

(c) EmoSet with $G = 12$      (d) EmoSet with $G = 8$

Figure 5: **Training and testing accuracy during the training process**. DAPO fails to conduct complete training due to dynamic sampling failure in EmoSet scenario. Refer to Sec. 4.3.

both in-domain, *e.g.*, Geo3K and out-of-domain, *e.g.*, MathVision, MathVista, and MathVerse, showing strong generalization performance under different rollout numbers. With $G = 12$, it achieves the highest overall accuracies (51.26 on math and 66.77 on emotion), and even with $G = 8$, it maintains superior results (50.80 and 66.55), validating that its mixed advantage formulation effectively mitigates advantage reversion and mirror issues while ensuring reliable optimization. Overall, it advances state-of-the-art performance with consistent gains across diverse reasoning tasks.

## 5 CONCLUSION

In our work, we focus on Group Relative Policy Optimization (GRPO) and observe that the advantage function plays a crucial role in evaluating trajectory importance. However, existing advantage formulations face two challenges: advantage reversion and advantage mirror. To address these issues, we propose Mixed Advantage Policy Optimization (MAPO). In particular, we uncover the trajectory certainty property and introduce advantage percent deviation for high-certainty trajectories. Furthermore, we dynamically reweight the advantage function according to trajectory certainty, thereby adaptively tailoring the advantage to sample-specific characteristics. Our method offers three key advantages: First, *No Architecture Dependency*: MAPO operates without additional model architectures, ensuring high transferability across different architectures. Second, *No Thinking Pattern Conflict*: our approach directly evaluates trajectory advantages while maintaining compatibility with diverse reasoning formats. Third, *No Hyper-Parameter Configuration*: by leveraging trajectory certainty to adaptively reweight sample advantage formulations, our method avoids the need for additional hyperparameters, thereby improving reinforcement effectiveness. MAPO has been validated across diverse scenarios, underscoring its potential for broader applications.

REPRODUCIBILITY

To facilitate the reproducibility, we provide the source code for both the training and evaluating framework in the supplementary material. All experimental settings, including key hyperparameter for training, are detailed in appendix B.2. We experiment on 8 NVIDIA A100 GPUs.

THE USE OF LARGE LANGUAGE MODELS (LLMS)

We disclose using LLMs solely to aid and polish writing—enhancing academic expression accuracy, argument coherence, and text logic. All core ideas, research designs, experiments, and conclusions are independently developed by the authors.

ETHICS AND SOCIAL IMPACT

This work is purely methodological, focusing on enhancing foundation model reasoning through Mixed Advantage Policy Optimization. It uses only public benchmark datasets and involves no personal data, or sensitive content. The method does not create or process private data, nor is it deployed in real-world applications. While stronger reasoning may indirectly influence downstream uses, our study does not explore deployment, bias, or misuse. The research is academic in purpose and poses no direct ethical or societal risks, aligning with responsible and trustworthy AI development.

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

APPENDIX

## A  NOTATION AND ALGORITHM

We provide the notation table in Tab. 4 and proposed method algorithm in algorithm 1.

---

**Algorithm 1:** MAPO

---

**Input:** Reference model $\pi_{ref}$, old model $\pi_{old}$, current policy model $\pi_\theta$, group size $G$, Training Step $E$,
      Current Step $e$, Training Batch $B$, Question Distribution $\rho_Q$, Query Prompt $q$

Initialize $\pi_\theta \leftarrow \pi_{ref}$

**for** $e = 1, 2, ..., E$ **do**
    $\pi_{old} \leftarrow \pi_\theta$
    $q \sim \rho_Q, o = \{o_i\}_{i=1}^G \sim \pi_{old}(\cdot|q)$ ;         `// Sample prompt with G trajectory.`
    $\boldsymbol{r} = \{r_i\}_{i=1}^G = R(o)$ ;                   `// Measure trajectory reward.`
    $\mu = \frac{1}{G}\sum_{r_i \in \boldsymbol{r}} r_i, \sigma = \sqrt{\frac{1}{G}\sum_{i=1}^G (r_i - \mu)^2}$ ;     `// Calculate static information.`
    *Advantage Percent Deviation*;
    $\hat{A}_i \leftarrow (\boldsymbol{r}, \mu, \sigma)$ via eq. (2), $\hat{A}_i^{APD} \leftarrow (\boldsymbol{r}, \mu)$ via eq. (4) ;     `// Measure advantage.`
    *Trajectory Certainty Reweight*;
    $N = \sum_{i=1}^G \mathbf{1}_{\{r_i=1\}}, p = \frac{N}{G}$ ;         `// Calculate trajectory success ratio.`
    $\lambda(p) = 1 - 4p(1-p)$ ;            `// Measure trajectory maturity degree.`
    $\hat{A}_i^* \leftarrow (\hat{A}_i, \hat{A}_i^{APD}, \lambda)$ through eq. (6) ;         `// Mixed advantage.`
    $\mathcal{J}_{GRPO}(\theta) \leftarrow (\hat{A}_i^*, o, \pi_\theta, \pi_{old}, \pi_{ref}), \theta = \theta - \eta \nabla \mathcal{J}_{GRPO}(\theta)$ ;     `// Update Weight.`
**end**

---

Table 4: **Notation used in MAPO**. Summary of key variables and operations in our method. The *Definition* column indicates where each symbol first appears in the main text.

| Symbol | Description | Definition |
|---|---|---|
| $q$ | Query prompt sampled from distribution $\rho_Q$ | eq. (1) |
| $o_i$ | $i$-th trajectory (rollout) sampled from $\pi_{\text{old}}$ | eq. (1) |
| $r_i$ | Reward assigned to trajectory $o_i$ | eq. (1) |
| $G$ | Group size (number of rollouts per query) | eq. (1) |
| $\pi_\theta, \pi_{\text{old}}, \pi_{\text{ref}}$ | Current, old, and reference policy models | eq. (1) |
| $J_{\text{GRPO}}(\theta)$ | Group Relative Policy Optimization objective | eq. (1) |
| $\beta$ | KL regularization coefficient | eq. (1) |
| $f_\epsilon(x, y)$ | Clipping function $\min(xy, \text{clip}(x, 1-\epsilon, 1+\epsilon)y)$ | eq. (1) |
| $\mathbb{D}_{KL}[\pi_\theta \| \pi_{\text{ref}}]$ | KL divergence between policy and reference model | eq. (1) |
| $R(q, o_i)$ | Reward function | Sec. 3.1 |
| $\mu, \sigma$ | Mean and standard deviation of rewards in group | eq. (2) |
| $\hat{A}_i$ | Standardized advantage $\frac{r_i - \mu}{\sigma}$ | eq. (2) |
| $N$ | Number of successful trajectories in a group | eq. (3) |
| $p$ | Empirical success ratio $p = \frac{N}{G}$ | eq. (3) |
| $\hat{A}_i^{\text{APD}}$ | Advantage Percent Deviation $\frac{r_i - \mu}{\mu + \epsilon}$ | eq. (4) |
| $\lambda(p)$ | Trajectory maturity degree $1 - 4p(1-p)$ | eq. (5) |
| $\hat{A}_i^*$ | Mixed advantage combining $\hat{A}_i$ and $\hat{A}_i^{\text{APD}}$ | eq. (6) |
| $r_{\text{Format}}, r_{\text{Accuracy}}$ | Format reward and accuracy reward | fig. 3 |
| $\varrho(p)$ | Gradient ratio $\nabla_\theta J_{\text{MAPO}} / \nabla_\theta J_{\text{GRPO}}$ | eq. (7) |
| $\mathcal{S}$ | Training distribution | Sec. 4.1 |
| $\mathcal{T} = \{T_t\}_{t=1}^{|\mathcal{T}|}$ | Set of unseen test datasets | Sec. 4.1 |
| $|\mathcal{T}|$ | Number of unseen test datasets | Sec. 4.1 |
| $\mathcal{A}^S, \mathcal{A}^T, \bar{\mathcal{A}}$ | In-domain, out-of-domain and average accuracy | Sec. 4.1 |

## B  EXPERIMENTAL INFORMATAION

### B.1  DATASET INTRODUCTION

We use the following two datasets from the mathematics and emotional tasks for experiments.

- 📝 Geo3K [arXiv'21] Lu et al. (2021) Designed for geometry problem solving, this dataset contains images, text, and formulas that require models to perform joint visual–symbolic reasoning.
- 😄 EmoSet [ICCV'23] Yang et al. (2023) This large-scale collection targets visual emotion recognition, covering diverse scenes and a broad range of emotion categories.

Furthermore, for above two scenarios, we respectively conduct the evaluation on the following out-of-domain datasets to vaildate its generalization ability.

- 📝 MathVista [arXiv'23] Lu et al. (2023) Serving as a benchmark for visual mathematical reasoning, spanning algebra, geometry, and word problems for cross-domain generalization.
- 📝 MathVision [NeurIPS'24] Wang et al. (2024) Proposed for multimodal mathematical reasoning, it emphasizes inference across visual diagrams and natural language expressions.
- 📝 MathVerse [ECCV'24] Zhang et al. (2024b) Built to assess model understanding of complex charts, geometric figures, and formula-rich inputs, emphasizing visual interpretation in reasoning.
- 😄 WEBEmo [ECCV'18] Panda et al. (2018) Comprising millions of web images, this dataset spans 7 high-level emotion categories and supports recognition and cross-domain emotion analysis.
- 😄 Emotion6 [CVPR'15] Peng et al. (2015) A classic benchmark for visual emotion recognition, consisting of 6 basic emotion categories and widely used for standard evaluation.

We plot a detailed dataset case illustration in fig. 6.

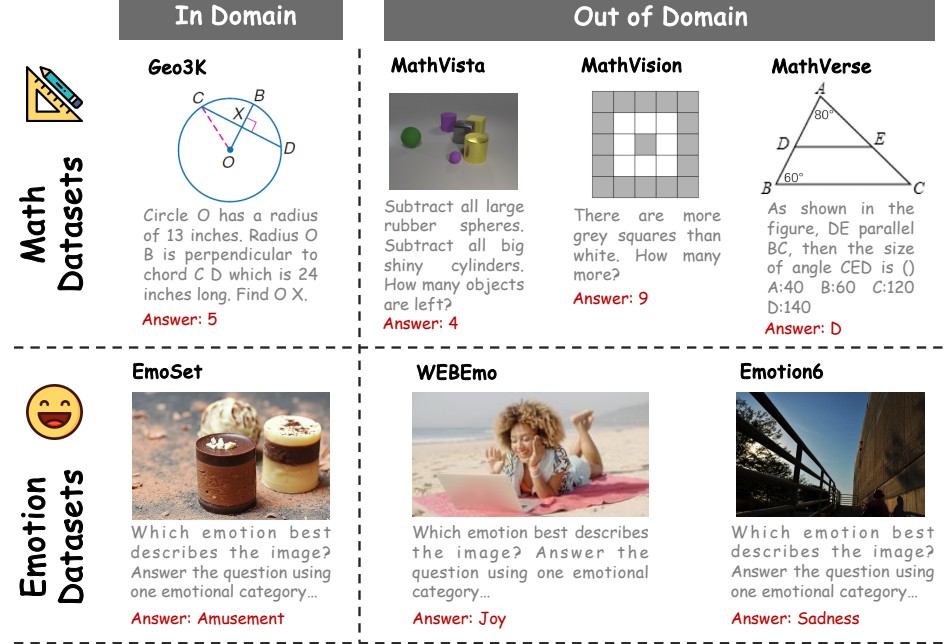

Figure 6: **Data Visualization** from the In Domain and Out of Domain datasets used in our experiments.

## B.2 IMPLEMENTATION DETAILS

We conduct the experiments on the EasyR1[1] Zheng et al. (2025) as our reinforcement learning training framework, which is built on Verl[2] Sheng et al. (2025). The rollout batch size is set to 512, and the global batch size is 128. The rollout temperature during training is fixed at 1.0, with Top-$p$ set to 0.99. To mitigate token-length bias, we compute the policy loss using a token-mean aggregation strategy. The vision tower of Qwen2.5-VL-7B is fine-tuned without freezing. The optimizer is AdamW Kingma & Ba (2014) with a learning rate of $1 \times 10^{-6}$, and the KL coefficient $\beta$ is set to $1 \times 10^{-2}$. For the validation setting, we set the temperature to 0.5. The maximum number of tokens to generate is 2048, and Top-$p$ sampling is 0.95. The training epoch is respectively set as $T = 20$ and $T = 15$ for Geo3K and EmoSet. Consequently, the training step is $E = 80$ for Geo3K and $E = 60$ EmoSet. These configurations are consistent with the EasyR1.

---

[1]https://github.com/hiyouga/EasyR1

[2]https://github.com/volcengine/verl

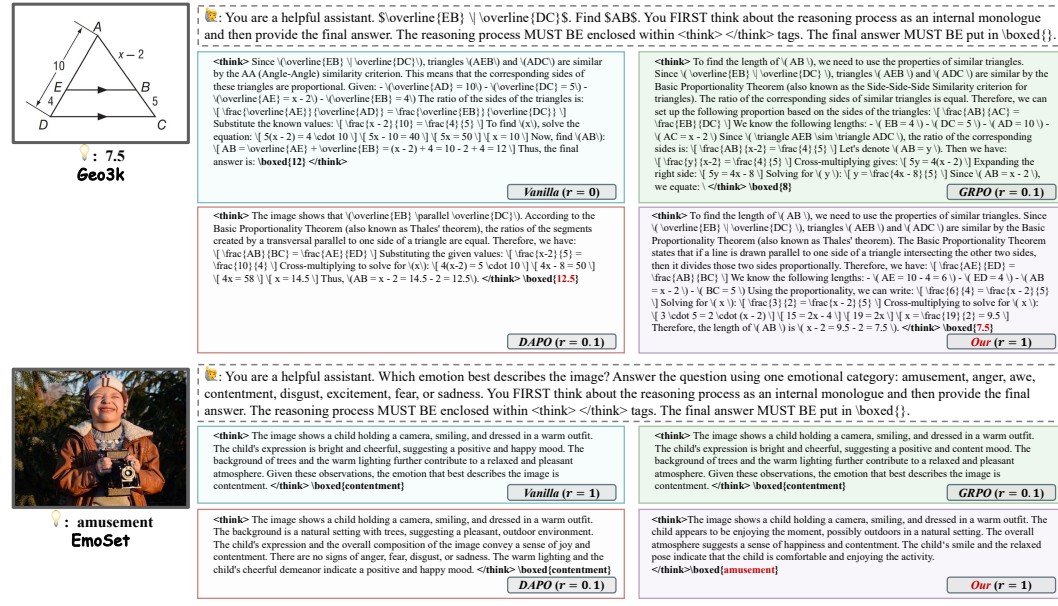

Figure 7: **In-Domain Case visualization**.

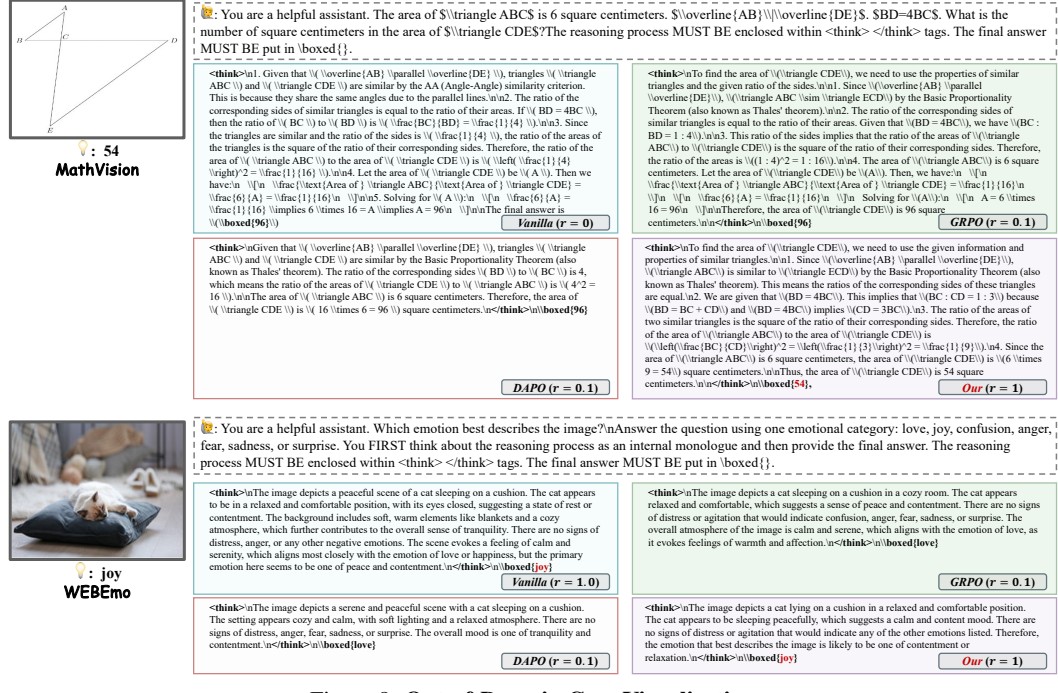

Figure 8: **Out-of-Domain Case Visualization**.

### B.3 VISUALIZATION ANALYSIS

We present the output cases for both in-domain (fig. 7) and out-of-domain distributions (fig. 8). For samples with high certainty, the existing GRPO leads to abnormal behavior. Both GRPO and DAPO exhibit degradation on the in-domain dataset EmoSet and the out-of-domain dataset WEBEmo in cases where the vanilla Qwen2.5-VL-7B-Instructversion could correctly answer. This suggests that the existing advantage distorts the optimization direction for samples with high trajectory certainty, ultimately leading to a performance decrease.

## C    THEORETICAL ANALYSIS

To better understand how proposed MAPO reshapes the optimization dynamics compared with GRPO, we provide a gradient-level analysis. *Without loss of generality*, we simplify the gradient analysis by ignoring clipping and KL regularization and modeling the reward as accuracy, i.e., a Bernoulli variable. For a prompt with $G$ rollouts and Bernoulli rewards $R_i \in \{0, 1\}$, define $A_i = R_i - \mu$ and let $p = \frac{N}{G}$, where $N = \sum_{i=1}^{G} \mathbf{1}_{\{R_i=1\}}$. Then $\mu = \frac{1}{G} \sum_i R_i = p$ and $\sigma = \sqrt{p(1-p)}$. Ignoring clipping and KL modules, the gradient of the objective is

$$\nabla_\theta \mathcal{J} = \mathbb{E}\Big[ \sum_{i,t} r_{i,t} \, \hat{A}_i \, \nabla_\theta \log \pi_\theta(a_{i,t} \mid s_{i,t}) \Big], \quad r_{i,t} = \frac{\pi_\theta(a_{i,t} \mid s_{i,t})}{\pi_{\text{old}}(a_{i,t} \mid s_{i,t})}. \tag{9}$$

For GRPO, the advantage is $\hat{A}_i^{\mathrm{G}} = A_i/\sigma$.

For MAPO,

$$\hat{A}_i^{\mathrm{M}} = (1 - \lambda(p))\frac{A_i}{\sigma} + \lambda(p)\frac{A_i}{p}, \qquad \lambda(p) = 1 - 4p(1-p). \tag{10}$$

Hence, for any trajectory, we define the ratio of the gradient as:

$$\boxed{\varrho(p) \triangleq \frac{\nabla_\theta \mathcal{J}_{\mathrm{MAPO}}}{\nabla_\theta \mathcal{J}_{\mathrm{GRPO}}} = \frac{\hat{A}_i^{\mathrm{M}}}{\hat{A}_i^{\mathrm{G}}} = (1 - \lambda(p)) + \lambda(p)\frac{\sigma}{p} = (1 - \lambda(p)) + \lambda(p)h(p)} \tag{11}$$

where $h(p) = \sqrt{\frac{1-p}{p}}$.

Next, we analyze the property of $\varrho(p)$. Since $h$ is smooth on $(0, 1)$ with

$$h'(p) = -\frac{1}{2p^2 \, h(p)} = -\frac{1}{2 \, p^{3/2}\sqrt{1-p}} < 0, \tag{12}$$

the derivative of $\varrho$ is

$$\varrho'(p) = 4(1 - 2p)\big(1 - h(p)\big) + \big(1 - 4p(1-p)\big) h'(p). \tag{13}$$

For $p \in (0, 1)$, we obtain that $\varrho'(p) \le 0$, and $\varrho(\frac{1}{2}) = 1$. Thus, we have:

$$\begin{cases} \varrho(p) > 1, & p \in (0, \frac{1}{2}), \\ \varrho(p) = 1, & p = \frac{1}{2}, \\ 0 < \varrho(p) < 1, & p \in (\frac{1}{2}, 1). \end{cases} \tag{14}$$

Which implies that MAPO leads to amplified gradients than GRPO on harder samples (with $p < \frac{1}{2}$), and smaller updates on easier samples (with $p > \frac{1}{2}$). This leads to the conclusion in Sec. 3.3.

