# OpenReview forum: "MAPO: MIXED ADVANTAGE POLICY OPTIMIZATION"
_ICLR.cc/2026/Conference — ICLR 2026 Conference Withdrawn Submission_

### Official Review · Reviewer_GUpS · 2025-10-24

**Soundness:** 1
**Presentation:** 1
**Contribution:** 2
**Rating:** 2
**Confidence:** 4

**Summary:**

The present work argues that the GRPO advantage estimator can suffer from two problems: (1) advantage reversion, where small standard deviation in the group rewards can inflate the advantages, and (2) advantage mirror, where the advantages are invariant to shifting. The authors then proposed Mixed Advantage Policy Optimization (MAPO) that combines a different advantage estimator that normalizes the advantages based on the mean, and a reweighting scheme that dynamically chooses the estimator based on the certainty of the group rewards. Experiments show marginal improvements over GRPO, suggesting its usefulness in RL reasoning tasks.

**Strengths:**

The idea of incorporating uncertainty into the policy optimization objective is an interesting idea, and the empirical results seem to suggest that such interventions may help GRPO-style training.

**Weaknesses:**

One of the major weaknesses of the paper is that the main claims (i.e., advantage reversion/mirror are problematic in GRPO training) are neither supported by theoretical justification nor empirical evidence beyond the intuitions given in the introduction. This makes it difficult to assess the effectiveness of the proposed fixes aside from the marginal gains demonstrated in the ablation studies. Furthermore, recent studies [1] have highlighted the importance of reporting uncertainties for assessing statistical significance, especially in the case of RL with large models. I believe this is particularly relevant for this work considering the marginal gains reported.

In addition, the paper is poorly organized (e.g., why is the discussion (sec 3.3) placed before the experiments rather than after?) The figures are also unclear (e.g., in Figure 4 (right) the axes labels are missing). It is also not clear to me why the experiments in sec 3.3 and sec 4 are using different baselines.

[1] Hochlehnert et al., "A Sober Look at Progress in Language Model Reasoning: Pitfalls and Paths to Reproducibility", COLM 2025

**Questions:**

1. Can the authors provide theoretical arguments on why the proposed modification is effective (e.g., convergence, bias-variance tradeoff)?
2. Why are the methods presented in Figure 4 not used in the experiments in Section 4?
3. In Table 2, what do bold and underline mean? In Table 3, why not use bold and underline for every task?
4. Can the authors explain why DAPO failed for EmoSet?

---

### Official Review · Reviewer_DCJ6 · 2025-11-01

**Soundness:** 3
**Presentation:** 3
**Contribution:** 2
**Rating:** 4
**Confidence:** 4

**Summary:**

The paper addresses a real stability issue with a neat two-term advantage and a smooth certainty gate, leading to reliable (but small) gains at negligible cost. However, the exclusive use of Qwen2.5-VL as the backbone (amid contamination concerns in math) and the fixed, non-learned adaptivity limit the conclusiveness and novelty. Adding leakage-resistant evals and diversified backbones, plus exploring learned/alternative gates, would substantially strengthen the paper.

**Strengths:**

- The paper cleanly isolates a real issue in GRPO-style training: z-score advantages behave poorly across groups with different trajectory “certainty.” The proposed two-term scheme (APD + certainty-weighted mixing) is easy to implement and numerically stable.
- Across math (MathVista/MathVision/MathVerse) and emotion (WEBEmo/Emotion6) OOD sets, the method delivers steady improvements over Vanilla/GRPO/DAPO, with ablations that justify using both components (APD and the certainty-based mixing).
- The method reuses the same rollout recipe; it only changes group-wise statistics and mixes two advantage measures. Wall-clock/compute should be essentially unchanged relative to GRPO.

**Weaknesses:**

- All results are on **Qwen2.5-VL-7B-Instruct**. This family has been audited for contamination on math benchmarks (https://arxiv.org/pdf/2507.10532); even though the paper evaluates on multimodal math (not exactly the same test suites), relying on a single possibly contaminated backbone weakens the claim of general gains. The paper would be stronger with additional backbones that are not implicated in leakage.
- The adaptivity is a fixed analytic gate $\lambda(p)$ rather than a learned/conditional schedule. Reported improvements are modest (often ~0.5–1.4 points). The paper does not test alternative certainty proxies or learned mixers, leaving open whether the fixed curve is near-optimal.
- No experiments on other strong open backbones (e.g., Llama-3.*-Vision, Gemma-3-Vision, InternVL-3.5). The emotion scenario also uses a single training set before OOD testing; broader coverage would improve external validity.

**Questions:**

1. Can you reproduce the main tables on at least two additional backbones (e.g., Llama-3.1-Vision, Gemma-3-Vision, InternVL-3.5) to decouple MAPO’s gains from Qwen-specific artifacts?
2. Will you include a math suite explicitly curated to avoid contamination (or a de-duplicated variant) to demonstrate robustness of the claimed improvements?
3. What is the rationale for fixing $\lambda(p)=1-4\,p(1-p)$ instead of learning $\lambda$ from $\{p,\mu,\sigma\}$ or using other certainty proxies (e.g., bootstrap CIs, reward-histogram entropy)? Any preliminary results?
4. Please provide sensitivity curves for the number of rollouts per prompt ($(G\in\{4,8,12,16\}$) and for each reward channel (accuracy vs. format vs. mixed).
5. Please confirm training steps, batch sizes, rollouts, and context lengths are matched across MAPO/GRPO/DAPO, and report any wall-clock overhead (expected to be negligible).
6. What exact rules are applied when the group mean in APD is near zero, or when $p$ is extreme? Do you clip APD or $\lambda$, or back off to a single advantage? Please include the code-level conditions.

---

### Official Review · Reviewer_C8s4 · 2025-11-01

**Soundness:** 1
**Presentation:** 2
**Contribution:** 2
**Rating:** 2
**Confidence:** 3

**Summary:**

In the context of RL for foundation models, the GRPO advantage function encounters both advantage reversion and advantage mirror problems which hinder allocation across different query samples (i.e., because the advantage is fixed throughout training). These issues motivate the authors to ask:

1. how to design the advantage function for high-certainty samples?

2. how to adaptively combine advantage functions for samples with varying trajectory certainty?

To answer these questions the authors correspondingly propose (1.) the Advantage Percent Deviation (APD) which essentially replaces the standard GRPO advantage $\hat{A}_i=\frac{r_i - \mu}{\sigma}$ with a mean-relative normalisation s.t. $\hat{A}^{\text{APD}}_i=\frac{r_i - \mu}{\mu}$, and (2.) Trajectory Certainty Reweight (TCR) which determines the sample advantage function based on Bernoilli modelling, dynamically weighting the advantage between $\hat{A}_i$ and $\hat{A}^{\text{APD}}_i$. The Mixed Advantage Policy Optimization (MAPO) method applies APD and TCR to adaptively adjust the advantage function during training based on trajectory certainty.

MAPO is evaluated using both mathematics (Geo3K, MathVista, MathVision, MathVerse) and emotion(EmoSet, WEBEmo, Emotion6) datasets, the Qwen2.5-VL-7B-Instruct base model, and GRPO and DAPO baselines for comparison. Ablation is used to evaluate $\hat{A}^{\text{APD}}_i$ in contrast to  $\hat{A}_i$, and an “out-of-domain validation” is also performed. Within domain, in Geo3K and EmoSet, MAPO provides enhanced scores over the baselines. Out-of-domain (Math-Vista,Vision,Verse and Emotion6) the results are more ambiguous.

**Strengths:**

I find the motivation and design of MAPO to be of interest, and potentially quite significant, if the analysis and evaluation can be substantially improved to support the claims made in the paper. The authors identify and address advantage reversion and advantage mirror problems that may arise given the standard advantage formation of GRPO. They address these challenges with an adaptive weighted advantage structure based on trajectory certainty and show some results that support improved scores in mathematics and emotional tasks.

**Weaknesses:**

Broadly speaking the quality of writing, and particularly grammar, in this paper is below the standard I would expect for ICLR. The Tables and Figures also have room for improvement e.g., Figure 4 and 5 lack axis labels, poorly legible legend in Figure 5, underline used in EMotion6 column of Table 3

In terms of technical content, whilst I find the motivation and the design of MAPO to be of-interest and potentially significant; and I think the ablation result is a step in the right direction, I find both the evaluation and discussion of the findings to fall short of the ICLR standard. The following important aspects are seemingly unaddressed and prevent me from understanding whether MAPO is functioning according to its design principles –

- In Figure 5, MAPO shows relatively convincing training accuracy improvements but this doesn’t seem to translate to testing accuracy.

- MAPO is outperformed by GRPO and DAPO in certain “out-of-domain” tasks e.g., MathVista and MathVerse.

- The above seems correlated on the rollout size G which MAPO depends upon for uncertainty estimation. It is unclear how performance and stability are impacted by G.

- It remains unclear how efficient and numerically stable MAPO is.

- No empirical consideration is given to the claim TCR “ensures that sample-specific characteristics are preserved, leading to a more faithful and stable evaluation of trajectory quality.” e.g., does certainty-based weighting moderate advantage in the way we hope?

- To the best of my knowledge there is no variance reporting in the evaluation process, nor any indication that more than one run was done. Tt’s not clear whether the reported improvements are statistically significant. It would be great if the authors could shed some light on this.


Writing lacking clarity or questionable (not exhaustive):

- “RL serves as the key mechanism for unlocking the reasoning ability in various domains.“ – I think this is overstating the current evidence.

- I think the introduction dives into advantage reversion and mirroring without sufficiently contextualising the role of the advantage function in GRPO. A few extra sentences would significantly improve the accessibility of the paper introduction. e.g. approximately covering: GRPO samples a group of trajectories, each trajectory receives a reward, the advantage function determines how much reward each trajectory should be assigned.

- Line 137, “reasoning models are now widely deployed locally, drawing attention from the research community to the efficiency of long chain-of-thought generation for foundation models.” ([pdf](zotero://open-pdf/library/items/YASLZRUI?page=3)) This is non-sequitur/should be clarified.

- Line 138, “. And utilizing the reinforcement technique to empower foundation models”. Sentences shouldn’t begin with ‘And’.

- Make sure acronyms are used once they have been defined e.g., line 150-151 “Multimodal Large Language Model” ([pdf](zotero://open-pdf/library/items/YASLZRUI?page=3))

- The legends in Figure 5 are poorly legible - it took my longer than usual to understand which line was MAPO etc.

- It’s unclear where the data for Figure 1 comes from? How can it be reproduced?


Minor/typos/grammar:

- Line 116, LLM should be plural, as should MLLM line 123

**Questions:**

- What is your intuition or principled understanding of why MAPO shows relatively convincing training accuracy improvements but this doesn’t seem to translate as significantly to testing accuracy?

- Why do you think MAPO is outperformed by GRPO and DAPO in certain “out-of-domain” tasks e.g., MathVista and MathVerse?

- The above seems correlated on the rollout size G which MAPO depends upon for uncertainty estimation. How might performance and stability be impacted by G?

- How efficient and numerically stable is MAPO?

- You claim that TCR “ensures that sample-specific characteristics are preserved, leading to a more faithful and stable evaluation of trajectory quality.” Can you determine whether certainty-based weighting moderates advantage in the way you hope?

- Can you provide details regarding the statistical significance of your results?

---

### Official Review · Reviewer_dE5m · 2025-11-01

**Soundness:** 2
**Presentation:** 2
**Contribution:** 2
**Rating:** 2
**Confidence:** 4

**Summary:**

The paper studies the advantage function used in GRPO for reasoning LLMs and identifies two failure modes—“advantage reversion” and “advantage mirror”—that occur on high-certainty samples. To address them, it proposes Mixed Advantage Policy Optimization (MAPO). The mixed advantage improves stability across certainty regimes and yields consistent, albeit modest, gains over GRPO and DAPO on math and emotion benchmarks with Qwen2.5-VL-7B under different rollout counts.

**Strengths:**

The proposed definition of trajectory certainty is reasonable (though there is prior art), and the resulting mixed advantage is simple, easy to implement, and practical to apply.

The paper is clearly written, with readable figures and tables.

**Weaknesses:**

1. The paper partitions sample certainty at the trajectory level and reweights accordingly. The authors should discuss related work on trajectory-level certainty/entropy-driven reweighting and sampling:
- [1] Vanlioglu (2025) uses entropy-guided sequence weighting to schedule exploration/exploitation in RL-based LLM fine-tuning;
- [2] Liu et al. (AAMAS 2024) present a trajectory-level perspective that systematically analyzes how data sampling techniques affect policy learning in RL.

These are highly aligned in motivation with the paper’s ‘Trajectory Certainty Reweight’. Besides, [2] also discusses using trajectory quality as a metric; in comparison, does Trajectory Certainty Reweight offer stronger advantages?”

2. The set of baselines appears limited. Compared with GRPO variants such as Dr.GRPO/GPG/TreeRPO, the performance advantage seems not particularly pronounced.

3. The experiments are further limited by the lack of statistical significance analysis.

4. There are quite a few typos and minor presentation issues; the authors should carefully proofread. For example, the subfigure title in Figure 4 reads “Performance durning Training Step,” which should be “during,” etc. Also, the manuscript use the wrong citation format, maybe caused by misusing \citet and \citep.


【1】Vanlioglu, Abdullah. "Entropy-guided sequence weighting for efficient exploration in RL-based LLM fine-tuning." arXiv preprint arXiv:2503.22456 (2025).
【2】Liu, J., ... , 2024, May. A trajectory perspective on the role of data sampling techniques in offline reinforcement learning. In Proceedings of the 23rd International Conference on Autonomous Agents and Multiagent Systems (pp. 1229-1237).

**Questions:**

Does λ(p) jitter, and is smoothing necessary?

Is the method broadly applicable across models of different parameter scales?

---

### Note · Authors · 2025-11-12

I have read and agree with the venue's withdrawal policy on behalf of myself and my co-authors.